# Learning Dynamics from Noisy Measurements using Deep Learning with a Runge-Kutta Constraint

**Pawan Goyal**
Max Planck Institute for Dynamics
of Complex Technical Systems
Sandtorstraße 1,
39106, Magdeburg, Germany
`goyalp@mpi-magdeburg.mpg.de`

**Peter Benner**
Max Planck Institute for Dynamics
of Complex Technical Systems
Sandtorstraße 1,
39106, Magdeburg, Germany
`benner@mpi-magdeburg.mpg.de`

## Abstract

Measurement noise is an integral part while collecting data of physical processes. Thus, noise removal is necessary to draw conclusions from these data and is essential to construct dynamic models using these data. This work discusses a methodology for learning dynamic models using noisy measurements and simultaneously obtaining denoised data. In our methodology, the main innovation can be seen in integrating deep neural networks with a numerical integration method. Precisely, we aim at learning a neural network that implicitly represents the data and an additional neural network that models the vector fields of the dependent variables. We combine these two networks by enforcing the constraint that the data at the next time-step can be obtained by following a numerical integration scheme. The proposed framework to identify a model predicting the vector field is effective under noisy measurements and provides denoised data. We demonstrate the effectiveness of the proposed method to learn models using a differential equation and present a comparison with the neural ODE approach.

## 1 Introduction

Uncovering dynamic models explaining physical phenomena and dynamic behaviors has been an active research area for several decades, see, e.g., [1–3]. When a model describing the underlying dynamics is available, it can be used for several engineering studies such as process design or predictions. Conventional approaches based on physical laws and empirical knowledge are often used to derive dynamical models. With some prior assumptions such as linearity of dynamical systems, dictionary-based, symbolic regression, many developments have been made possible, see, [4–10]. However, their success depends on the accuracy of the hypotheses. Obtaining suitable hypotheses is impenetrable for many complex systems, e.g., understanding the Arctic ice pack dynamics, sea ice, power grids, neuroscience, or finance, to only name a few applications.

Machine learning techniques, particularly deep learning-based, have emerged as powerful methods capable of expressing any complex function in a black-box manner. Neural network-based approaches in the context of dynamical systems have been discussed in [11–14] decades ago. Neural-networks are incorporated in various ways in the course of learning models, e.g., for prediction, see [13, 15–27]. We particularly highlight the recent work on neural ordinary differential equation (ODE) [25] which has successfully been applied to learn dynamical models. Furthermore, data acquired using imaging devices or sensors are contaminated with measurement noise. Therefore, systematic approaches that learn a dynamic model with proper treatment of noise are required. The approaches mentioned above do not perform any specific noise treatment. The work in [28] proposes a framework that explicitly incorporates the noise into a numerical time-stepping method. Though the approach has

Workshop Paper at The Symbiosis of Deep Learning and Differential Equations Workshop at NeurIPS 2021.

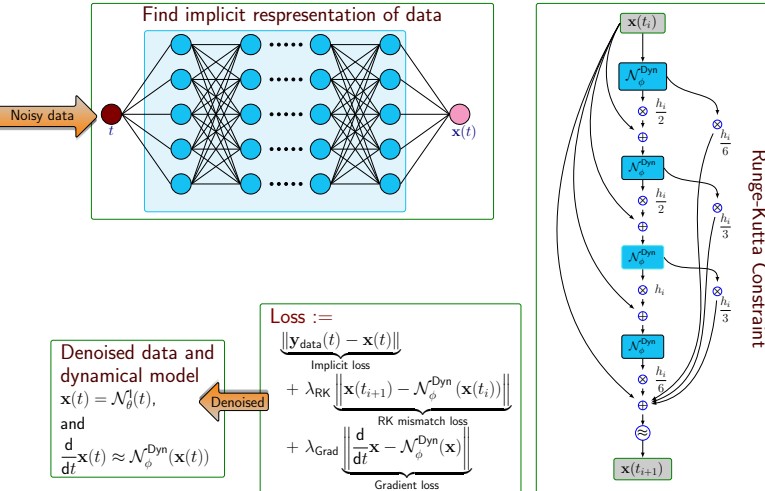

Figure 1: Illustration of the framework to denoise data and to learn a dynamical model. For this, we find an approximate implicit representation of the data by a network ($\mathcal{N}_\theta^{\mathsf{I}}$) and another network for the vector field ($\mathcal{N}_\phi^{\mathsf{Dyn}}$). These networks are connected by enforcing a Runge-Kutta scheme, shown in (c), on the output of the network $\mathcal{N}_\theta^{\mathsf{I}}$.

shown promising directions, its scalability remains ambiguous as the approach explicitly estimates noise and aims to decompose the signal explicitly into noise and ground truth.

**Our contributions:** Our work introduces a framework to learn dynamical models by innovatively blending deep learning with numerical integration methods from noisy and sparse measurements. Precisely, we aim at learning two networks; one that implicitly represents given measurement data and the second one approximates the vector field; we connect these two networks by enforcing a numerical integration scheme as depicted in Figure 1. The appeal of the approach is that we do not require an explicit noise estimate to learn a model. Furthermore, the approach is applicable even if the dependent variables are sampled on different time grids, which can also be irregular. We illustrate the promises of the proposed approach with an example of a cubic damped differential equation and present a comparison with the neural ODE approach [25]. Last but not least, the implicit representation of the time series data by a neural network allows to obtain gradient samples by backpropagation without the need for numerical approximation.

## 2 Learning Dynamical Models using Deep Learning Constraint by a Runge-Kutta Scheme

For methods to learn dynamical models, the quality of measurement data plays a significant role in ensuring the accuracy of the learned models. Before employing any data-driven method, de-noising the data is a vital step and is typically done using classical methods, e.g., smothering techniques or moving averages. Though they perform good in general to remove a large part of the noise, the result is still contaminated. In this section, we discuss our framework to learn dynamic models using noisy measurements without explicitly estimating the noise. To that end, we utilize the powerful approximation capabilities of deep neural networks and its automatic differentiation feature with a numerical integration scheme. In this work, we focus on the *fourth-order Runge-Kutta* (RK4) scheme; however, the framework is flexible to use any other numerical integration scheme or higher-order Runge-Kutta schemes. In fact, we can also make use of the neural ODE [25] framework to replace the RK4 scheme. For this, let us consider an autonomous nonlinear differential equation:

$$\frac{\mathrm{d}}{\mathrm{d}t}\mathbf{x}(t) = \mathbf{g}(\mathbf{x}(t)), \quad \mathbf{x}(0) = \mathbf{x}_0, \tag{1}$$

where $\mathbf{x}(t) \in \mathbb{R}^n$ denotes the solution at time $t$, and the continuous function $\mathbf{g}(\cdot) : \mathbb{R}^n \to \mathbb{R}^n$ defines the vector field. Given (1) and $\mathbf{x}_{t_i}$, we can predict $\mathbf{x}(t)$ at time $t_{i+1}$ using the RK4 scheme as follows:

$$\mathbf{x}(t_{i+1}) \approx \mathbf{x}(t_j) + h_i \left( \tfrac{1}{6}\mathbf{k}_1 + \tfrac{1}{3}\mathbf{k}_2 + \tfrac{1}{3}\mathbf{k}_3 + \tfrac{1}{6}\mathbf{k}_4 \right), \quad \text{with } h_i = t_{i+1} - t_i, \tag{2}$$

where

$$\mathbf{k}_1 = \mathbf{g}\left(\mathbf{x}(t_i)\right), \quad \mathbf{k}_2 = \mathbf{g}\left(\mathbf{x}(t_i) + \tfrac{h_i}{2}\mathbf{k}_1\right), \quad \mathbf{k}_3 = \mathbf{g}\left(\mathbf{x}(t_i) + \tfrac{h_i}{2}\mathbf{k}_2\right), \quad \mathbf{k}_4 = \mathbf{g}\left(\mathbf{x}(t_i) + h_i\mathbf{k}_3\right).$$

In this paper, we use the short-hand notation for (2), $\mathbf{x}(t_{i+1}) \approx \Pi_{\mathsf{RK}}(\mathbf{x}(t_i))$, and we assume that the state $\mathbf{x}(t)$ is fully observed.

Next, we discuss our framework to learn dynamical models from noisy measurements by blending deep neural networks with the RK4 scheme. The approach involves two networks. The first network implicitly represents the variable as shown in Figure 1, and the second network approximates the vector field, or the function $\mathbf{g}(\cdot)$. These two networks are connected by attenuating the RK4 constraints. That is, the output of the implicit network is not only in the vicinity of the measurement data but also approximately follows the RK4 scheme as depicted in Figure 1. For clarity, let us denote noisy measurement data at time $t_i$ by $\mathbf{y}(t_i)$. Furthermore, we consider a feed-forward neural network, denoted by $\mathcal{N}_\theta^{\mathsf{I}}$ parameterized by $\theta$, that approximates the measurement data, i.e.,

$$\mathcal{N}_\theta^{\mathsf{I}}(t_i) := \mathbf{x}(t_i) \approx \mathbf{y}(t_i), \quad i \in \{1, \ldots, m\}. \tag{3}$$

Additionally, let us denote another neural network by $\mathcal{N}_\phi^{\mathsf{Dyn}}$ parameterized by $\phi$ that approximates $\mathbf{g}(\cdot)$ in (1). We connect these two networks by enforcing the output of the network $\mathcal{N}_\theta^{\mathsf{I}}$ to respect the RK4 scheme, i.e.,

$$\mathbf{x}(t_{i+1}) \approx \Pi_{\mathsf{RK}}\mathbf{x}(t_i), \qquad \text{and} \quad \tfrac{\mathrm{d}}{\mathrm{d}t}\mathbf{x}(t_i) \approx \mathcal{N}_\phi^{\mathsf{Dyn}}(\mathbf{x}(t_i)). \tag{4}$$

As a result, our goal becomes to determine the network parameters $\{\theta, \phi\}$ such that the following loss is minimized:

$$\mathcal{L} = \lambda_{\mathsf{MSE}} \cdot \mathcal{L}_{\mathsf{MSE}} + \lambda_{\mathsf{RK}} \cdot \mathcal{L}_{\mathsf{RK}} + \lambda_{\mathsf{Grad}} \cdot \mathcal{L}_{\mathsf{Grad}}, \tag{5}$$

in which $\mathcal{L}_{\mathsf{MSE}} := \|\mathcal{N}_\theta^{\mathsf{I}}(t_i) - \mathbf{y}(t_i)\|^2$, where $\|\cdot\|$ represents the mean squared error. The term $\mathcal{L}_{\mathsf{RK}}$ links the two networks by the RK4 scheme. Precisely, the term $\mathcal{L}_{\mathsf{RK}}$ castigates the mismatch between $\mathbf{x}(t_{i+1})$ and $\Pi_{\mathsf{RK}}\mathbf{x}(t_i)$, i.e., $\|\mathbf{x}(t_{i+1}) - \Pi_{\mathsf{RK}}(\mathbf{x}(t_i))\|^2$. Moreover, the vector field at the output of the implicit network can be computed directly using automatic differentiation, but it also can be computed using the network $\mathcal{N}_\phi^{\mathsf{Dyn}}$. The term $\mathcal{L}_{\mathsf{Grad}}$ penalizes the mismatch, i.e., $\|\mathcal{N}_\phi^{\mathsf{Dyn}}(\mathbf{x}(t_i)) - \tfrac{\mathrm{d}}{\mathrm{d}t}\mathbf{x}(t_i)\|^2$. $\lambda_{\mathsf{MSE}}$, $\lambda_{\mathsf{RK}}$, and $\lambda_{\mathsf{Grad}}$ are the corresponding regularization parameters.

The total loss $\mathcal{L}$ can be minimized using a gradient-based optimizer such as Adam [29]. Once the networks are trained and have found their parameters that minimize the loss, we can generate the denoised variables using the implicit network $\mathcal{N}_\theta^{\mathsf{I}}$, and the vector field by the network $\mathcal{N}_\phi^{\mathsf{Dyn}}$. Note that due to the involvement of the implicit nature of the network, the measurement data can be at variable time steps; in fact, the dependent variables can be measured at a different time-grid as well, and we can estimate the solution at any arbitrary time. Moreover, we also obtain the network $\mathcal{N}_\phi^{\mathsf{Dyn}}(\cdot)$ that approximately provides the vector field for $\mathbf{x}$.

## 3   Numerical Experiments

We demonstrate the proposed methodology using a cubic damped oscillatory differential equation. We corrupt the data by adding mean-zero Gaussian white noise of variance $\{1\%, 5\%, 10\%, 20\%\}$. We aim to obtain a denoised signal and a model, defining its vector field. Before employing the method, we perform a pre-processing step to noisy data using a low-pass filter to remove a large portion of the high-frequency noise. We compare our methodology with the neural ODE framework [25], which also focuses on learning a neural network, defining the underlying vector field. The details about the neural architecture and training are given in the appendices.

Having trained both models for approximating the vector field, we compare the learned vector fields in Figure 2 and plot the mean and median of the vector field errors, see Figure 3 (left). We observe that the proposed approach with denoising and learning a dynamical model simultaneously outperforms neural ODE for noisy cases, which is quite apparent for large noise; see the last two rows of Figure 2. Moreover, in our preliminary experiments, we have observed that the proposed approach is approximately two times faster on both, CPU and GPU, see Figure 3 (right) and have a similar type of convergence with respect to the number of epochs. Moreover, we obtain the denoised data from our implicit network directly, which is compared with the truth data in Figure 5, which is not possible using neural ODE standalone.

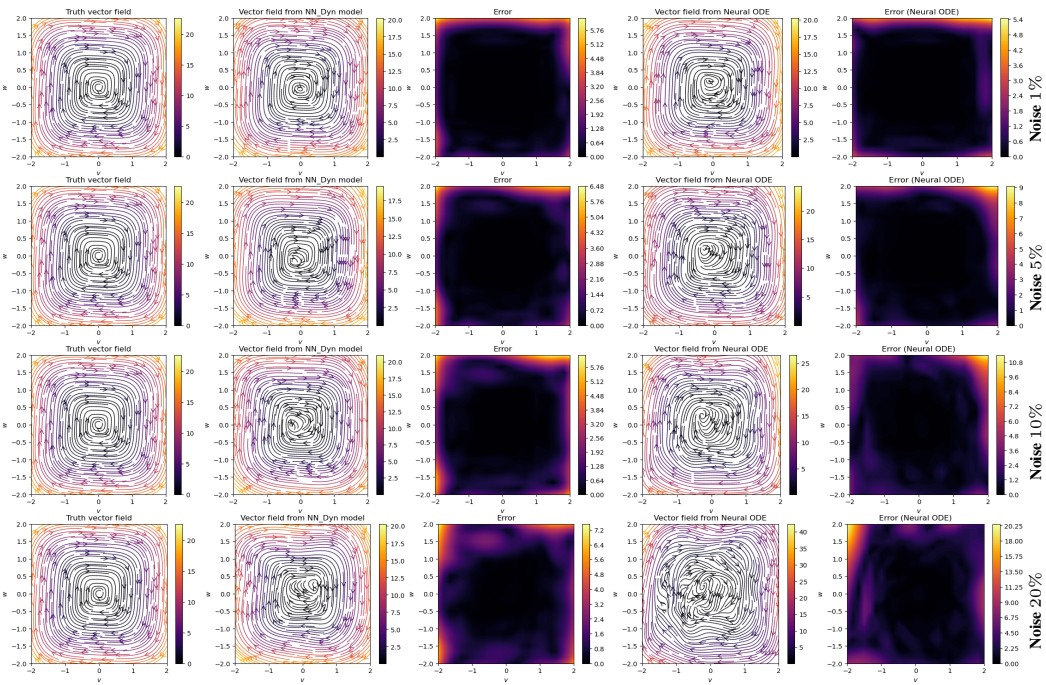

Figure 2: A comparison of vector fields of the ground-truth and learned models for various noise.

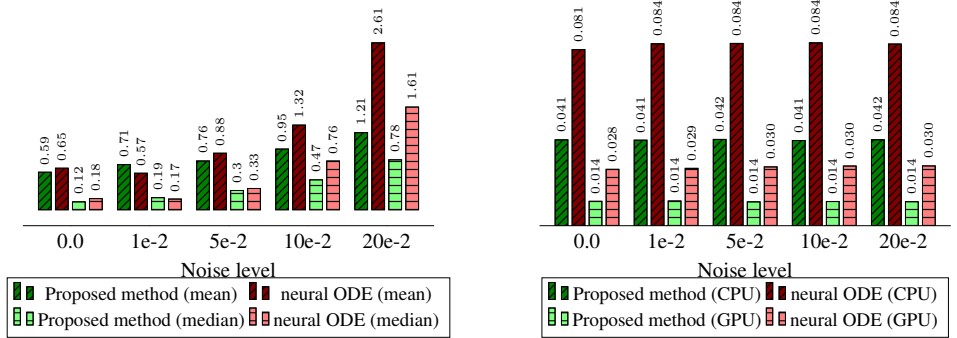

Figure 3: The left figure shows the mean and median errors between the truth and learned vector fields. The right figure indicates computational time per epoch on CPU and GPU.

# 4    Discussion

In this work, we have presented a new paradigm for learning dynamical models from noisy measurement data. Our framework blends universal approximation capabilities of deep neural networks with a numerical integration scheme, namely the fourth-order Runge-Kutta scheme. The proposed scheme involves two networks to learn an implicit representation of the measurement data and the vector field. These networks are combined by enforcing that the output of the implicit network respects the integration scheme. Our aim is to obtain denoised data as the output of the implicit network in the vicinity of noisy measurement such that a differential equation can define its evolution. The whole approach can be seen as neural ODE for noisy data if the Runge-Kutta scheme is replaced by an integral representation. Moreover, we emphasize that the proposed approach is applicable when the dependent data are not measured in the same time frame due to the involvement of the implicit network, where the applicability of the standalone neural ODE [25] would not be possible.

The proposed methodology opens various new directions for further research. It would be interesting to investigate the performance of the approach by replacing the Runge-Kutta scheme with an integral like done in neural ODE [25], which would allow predicting the trajectory for a longer time horizon

without committing a large integration error. Moreover, we still need to examine the efficiency of the proposed method when the measurement of the dependent variable is collected on different time-grids. In addition to these, in several cases, data are either partially observed or are high-dimensional. Hence, it would be worthwhile to combine the encoder-decoder idea to identify the latent space to represent dynamical systems. Last but not least, we need to perform a thorough computational study of the performance of the approach and assess efficient training under the circumstances mentioned above.

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

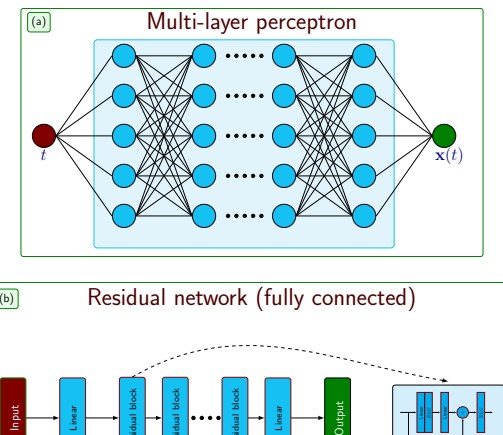

Figure 4: The figure shows two potential simple architectures that can be used to learn either implicit representation or to approximate the underlying vector field. Diagram (a) is a simple multi-layer perceptron, and (b) is a residual-type network but fully connected.

## A    Suitable Neural Networks Architectures

Here, we briefly discuss neural network architectures suitable for our proposed approach. We require two neural networks for our framework, one for learning the implicit representation $\mathcal{N}_\theta^{\mathsf{I}}$ and the second one $\mathcal{N}_\theta^{\mathsf{Dyn}}$ is to learn the vector field. For implicit representation, we use a fully connected multi-layer perceptron (MLP) as depicted in Figure 4(a) with periodic activation functions (e.g., $\sin$) [30] which has shown its ability to capture finely detailed features as well as the gradients of a function. To approximate the vector field, we consider a simple residual-type network as illustrated in Figure 4(b) with *exponential linear unit* (ELU) as an activation function [31]. We choose ELU as the activation function since it is continuous and differentiable and resembles a widely used activation function, namely rectified linear unit (ReLU).

## B    Cubic damped model

We consider a damped cubic system, which is described by

$$
\begin{aligned}
\dot{\mathbf{x}}(t) &= -0.1\mathbf{x}(t)^3 + 2.0\mathbf{y}(t)^3, \\
\dot{\mathbf{y}}(t) &= -2.0\mathbf{x}(t)^3 - 0.1\mathbf{y}(t)^3.
\end{aligned}
\tag{6}
$$

It has been one of the benchmark examples in discovering models using data, see, e.g., [32, 33] but there, it is assumed that the dynamics can be given sparsely in a high-dimensional feature dictionary. Here, we do not make any such assumption and instead learn the vector field using a neural network along the lines of [25, 34]. This example has been considered in [25] as well for learning neural ODE. Experiments are performed on Intel® Xeon® Silver 4110 CPU @ 2.10GHz for CPU computations, and on P100 Nvidia® for GPU computations.

For this example, we take $2\,500$ data points in the time interval $[0, 25]$ with $dt = 10^{-2}$ by simulating the model using the initial condition $[2, 0]$ as done in [25]. We add various levels of noise in the clean data to have noisy measurements synthetically – we corrupt data using mean-zero Gaussian white noise of standard deviation $\Sigma\%$. For preprocessing, we employ a low-pass filter of order 3 from the `scipy` library.

We construct neural networks for implicit representation and the vector field with the parameters given in Table 1. We train the networks using ADAM [29] and the PyTorch library [35] for $15,000$ epochs. For training neural ODE, we have used the `torchdiffeq` library [25], and have created 2498 sequences and integrating them into the time-span $dt$ to be in accordance with our one-time

| Example | Networks | Neurons | Layers or residual blocks | Learning rates |
|---|---|---|---|---|
| Cubic oscillator | For implicit representation | 20 | 4 | $5 \cdot 10^{-4}$ |
|  | For approximating vector field | 20 | 4 | $10^{-3}$ |

Table 1: The table shows the information about network architectures and learning rates.

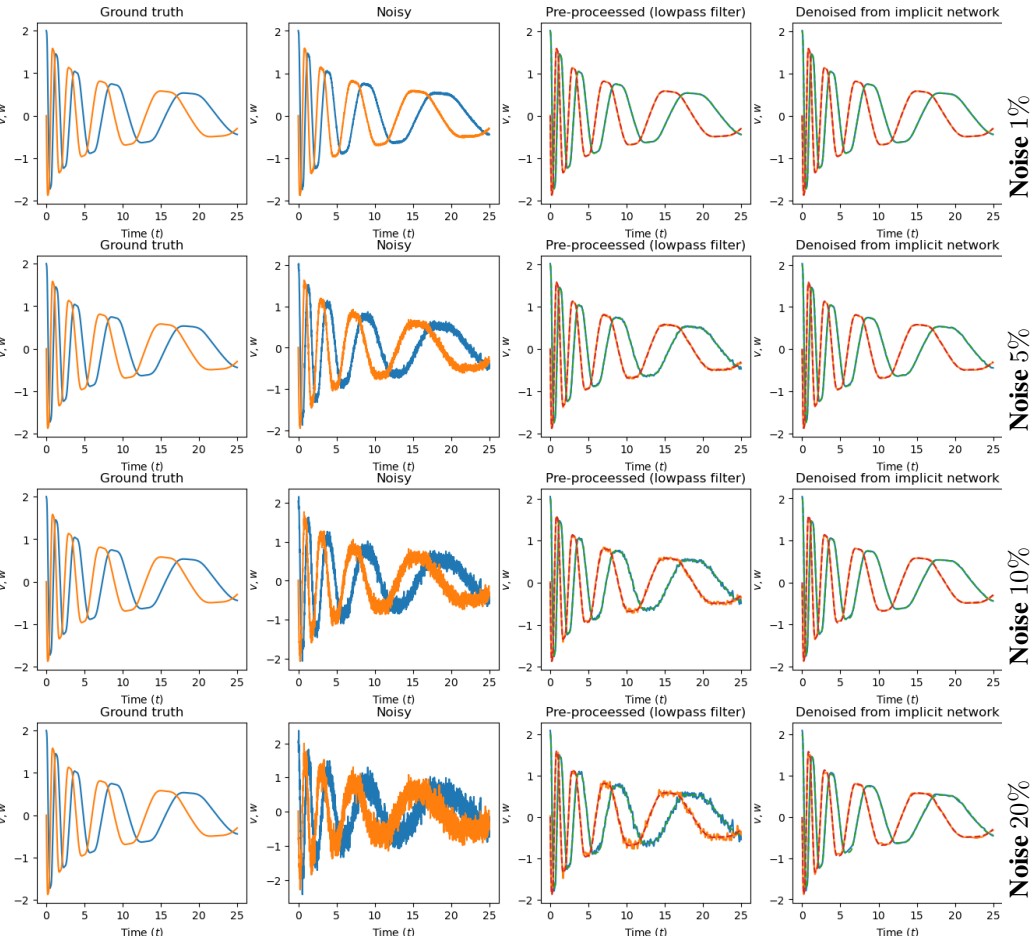

Figure 5: The figure shows the clean data, noisy, preprocessed, and denoised data from trained implicit network for various noise level.

ahead prediction in the Runge-Kutta scheme. We have also employed a scheduler that reduces the learning by one-tenth after each $5,000$ epochs.

For our approach as well, we have taken all data in the dataloader and train networks with parameters $\lambda_{\mathsf{MSE}} = 100.0$, $\lambda_{\mathsf{Dyn}} = 1.0$ and $\lambda_{\mathsf{Grad}} = 1.0$ in the loss function (5) for noise $\{1\%, 5\%\}$, and for $\{10\%, 20\%\}$, we reduced $\lambda_{\mathsf{MSE}}$ to $10.0$ to avoid over-fitting of noisy measurement data. Having trained networks, we have an implicit network to obtain denoised signal and a neural network approximating the vector field.

We plot the results in Figure 5, where we show noisy, preprocessed, and denoised data. We plot the streamlines of the vector field, obtained using the trained neural networks (our approach and neural ODE) in Figure 2 in the domain $[-2, 2] \times [-2, 2]$ by taking 25 points in each direction. It indicates a better performance of our approach compared to neural ODE for the more noisy case. Moreover, we plot the progress of the losses in (5) with respect to epochs in Figure 6 for different noise levels. We empirically find that the data fidelity loss is more for higher noise as one can anticipate, whereas the other two losses are more or less of the same order for various noise. Therefore, we have seen

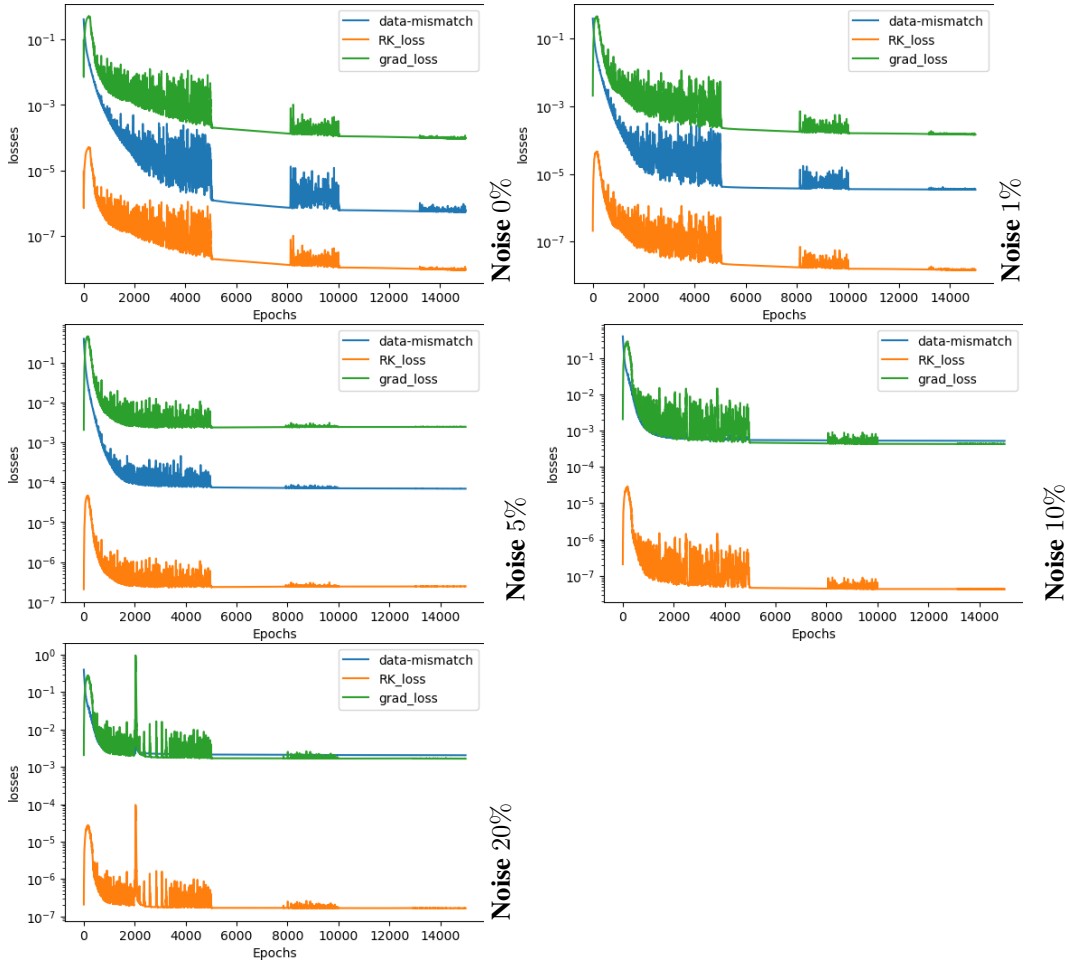

Figure 6: The figures indicate the losses $\mathcal{L}_{\mathsf{MSE}}$, $\mathcal{L}_{\mathsf{RK}}$ and $\mathcal{L}_{\mathsf{Grad}}$ in (5) with respect to epoch training and different level of noise.

a better performance of our approach as the trained implicit network aims at generated data whose trajectory can be explained by the neural ODE.

