# OpenReview forum: "Learning Dynamics from Noisy Measurements using Deep Learning with a Runge-Kutta Constraint"
_NeurIPS.cc/2021/Workshop/DLDE — DLDE Workshop -- NeurIPS 2021 Poster_

### Official Review · Reviewer_pFbC · 2021-10-04

**Confidence:** 3

**Review:**

## Summary

This paper combines neural ODEs with physics-informed neural networks. The former parameterises the vector field of an ODE by a neural network; the latter approximates the solution of an ODE by a neural network.

Supposedly this is supposed help model dynamics for which only poor-quality (noisy) data have been observed, although no justification is offered for this claim.

## Problems

The advantage of combing these two approaches is simply not evidenced. Why does this approach yield robustness to noise? How do the results obtained differ -- practically speaking -- from those obtained by simply approximating the vector field with a neural network and training a neural ODE as usual? This is a central weakness of the paper.

The text repeats itself quite a lot -- the wording could do with being tightened substantially -- e.g. repeated discussion about measurement noise.

The term "implicit representation" is quite suggestive of either deep implicit models (https://implicit-layers-tutorial.org/) or implicit representations (https://arxiv.org/abs/2006.09661). The connections to either don't seem to be that strong, so I'm not a fan of using this terminology here.

Figure 1 is too dense/small to be read clearly.

The $\lambda_{RK}$ loss in Figure seems to be slightly misrepresented: a single application of just $\mathcal{N}^{Dyn}$ is used rather than the RK scheme.

The Runge--Kutta scheme used is one of two known as "RK4". The other, sometimes termed "RK4 with 3/8 rule", is generally more popular as achieving slightly better results. (And even that is generally not favoured -- the current "best" mid-order non-stiff RK scheme is generally held to be "tsit5" introduced by https://www.sciencedirect.com/science/article/pii/S0898122111004706.)

## Limitations

It would seem that the proposed scheme assumes that the dynamics follow a Markov property; equivalently that $x$ is fully observed. This seems like a limitation as in general only some channels might be observed.

(For this reason neural differential equations often consider the state $x$ as evolving in some latent space; a linear map is then used to transform $x$ into data space. This is precisely analogous to the typical way in which an RNN is used.)

I suspect in this context that this this limitation can be overcome. However it is common to see authors implicitly assume such Markov behaviour, without any comment, despite it really restricting the kind of behaviour that can be modelled.

**Score:**

2: Borderline paper

---

> ### Author Response · Authors · 2021-10-20
> **Response**
>
> ###### General comments
> First of all, we thank you for your detailed evaluation of the paper. Before we answer your questions/issues, we like to clarify a few points about our method:
>
> * Our method aims at not only learning the vector field but also at obtaining denoised data. Obtaining denoised data by applying neural ODE is not possible. The denoised data can then be used for the discovery of governing equations, e.g., by employing [Cranmer et al. NeurIPS 2020].
> * In case of poor-quality measurements, the implicit network in our method focuses on generating data that is close to measurement, and the evolution of the generated data can be explained by neural ODEs defined by a neural network (which also needs to be learned). This is the main motivation of the work.
> * Benefiting from the trained implicit network, our method is capable of handling data that are not measured at the same time grid. Elaborating on this, assume there are two variables $x$ and $y$, and $x$ is measured on time-grid $T := \{t_1,\ldots,t_n\}$, but $y$ is measured on the time grid $\tilde T := \{\tilde t_1,\ldots, \tilde t_m\}$ with $T \cap \tilde T = 0$. We can then estimate these points on the same time grid using the implicit network, which can then be fed into neural ODE. The straightforward application of neural ODE would be impossible in this case.
> * Furthermore, since measurements are of poor quality, the initial condition which is needed to train neural ODE is noisy, and a precise estimate is not known. This imposes a big constraint to employing neural ODE for poor-quality measurements.
> * Hence, we needed to couple with implicit networks that yield denoised data close to the measurement data, such that the trajectory of the denoised data can be given by an ODE defined by a neural network. With this aim, we have proposed a scheme that combines implicit network and ODE (vector field defined by a neural network). We have focused on an RK scheme that approximates an integral in a given time span. However, one has the freedom to choose any other integral scheme, e.g., 3/8 RK scheme, Euler method, high-order RK schemes, or an integral form itself using neural ODE.
>
> ###### Respond to the comments
> * Basically, we seek to find the denoised data whose dynamics can be given by a neural ODE. Naturally, continuous dynamical models do not define the dynamics of noisy measurements. Therefore, we require two networks -- one network that generates denoised data, and the second tries to model denoised data dynamics using neural ODE. Training both simultaneously yields good results. However, a deep investigation remains open why the proposed method performs robustly to noise. But our empirical studies suggest that the implicit network aims at generating the data whose dynamics can be given by a neural ODE in the vicinity of the measurement data. We now provide figures (see Figure 6 in the paper), which support our arguments.
>
> * In the revised version, we have compared with neural ODE (using \texttt{torchdiffeq} library). Before applying the method to noisy measurements, we even remove a large part of the noise from the data using a low-pass filter. We have observed better performance of the proposed approach compared to neural ODE in both computational efficiency and accuracy.
>
> * Our implicit representation terminology matches with the paper \url{https://arxiv.org/abs/2006.09661}. Basically, we also are learning a continuous function from data. Therefore, we prefer to stick to this notation in our paper.
>
> * We agree that there are more advanced methods in the literature. However, in our work, we have used the classical RK4 scheme for simplicity and also note that it is widely used to date. However, our framework is quite flexible, and one may incorporate any other scheme, including the continuous one proposed in the neural ODE. Moreover, we highlight that we do integration between two time-steps, and since our measurements are not exact, we can safely use the classical RK4 scheme, which has given us good results. In our future work, we would investigate how different integration (approximation) schemes affect the results and training of networks.
>
> ###### Respond to the limitations' comments
> * We agree that in our current framework, we assume to have full state observation. We now mention this in our article. But as you rightly pointed out, we can lift the observation to a latent space using an encoder and enforce neural ODE constraints in the latent space. It would follow the same procedure as proposed in [Chen et al., 2019]. Since the page limits are only 4, we exclude this discussion for our future investigation and have mentioned this in the discussion section. Additionally, we had mentioned in our discussion that it would also be interesting to look at low-dimensional manifold for high-dimensional data, coming from e.g., PDEs since many of the high-dimensional dynamical systems evolve in low-dimensional manifolds.

---

### Official Review · Reviewer_XKGu · 2021-10-11

**Confidence:** 2

**Review:**

The paper proposes an approach to learning dynamics in a noisy input setting. The method learns a neural representation of the noisy data to use in a neural net which learns the dynamics of the given system. Both these neural nets are 'connected' using a numerical integration scheme (here RK4).

**Comments**

The overall paper presentation is good and easy to understand. The approach to using 2 neural nets for implicitly denoising noisy data and using it to learn dynamics is interesting.

**Issues**

The initial discussion about measurement noise could be more concise.
Diagrams in Fig 1 and plots in Fig 2 are quite small, which makes interpreting them difficult.


**Score:**

3: Good paper

---

> ### Author Response · Authors · 2021-10-20
> **Response**
>
> Thank you for your positive remarks on the manuscript. We have made the paper and discussion more concise. Also, we have improved the figures bigger as well.
>
> We also would like to inform you that we have modified the numerical section substantially and have a comparison with a Neural ODE work [Chen et al. NeurIPS 2018]. It indicates a better performance of our approach as compared to Neural ODE for noisy measurements.

---

### Official Review · Reviewer_za1E · 2021-10-11

**Confidence:** 3

**Review:**

The authors present an approach to learning dynamic models from noisy measurements by combining deep learning with a classical numerical integration method. The method consists of two networks; one that implicitly represents given measurement data and the second one approximates the vector field; and a loss function that uses Runge-Kutta Constraint and forces the networks to achieve their goal.


The idea of combining the two approaches is interesting and could open the door to further work, but in my opinion the method can be better justified. In particular, the experiments presented in the work describe how the method deals with different problems, but no comparison is made with the classical methods in terms of computational cost or quality of the solution. Even comparing success (error versus effort) within the components of the method can give intuition to its justification (e.g., regularization coefficients $\lambda_{\sf RK}, \lambda_{ \sf Grad}$, CNN vs. RNN, network size etc.).

**Score:**

3: Good paper

---

> ### Author Response · Authors · 2021-10-20
> **Response**
>
> Thank you for your positive response. We have done several amendments to the paper. We have also performed a comparison study with Neural ODE work which is one of the state-of-the-art methods to learn models using neural networks.  We have noticed a better performance both computationally and accuracy-wise. An intuition to explain why our method performs better is that we aim at learning data in the vicinity of the measurement using implicit networks. If we employ directly neural ODE, we observe that it does not perform well due to noisy initial conditions which need to be fed while training neural ODE. Please see the updated numerical results. We present only one example due to limited space and have removed other examples e.g. for PDEs.
>
> * The	regularization coefficients $\lambda_{RK}, \lambda_{Grad}$ and $\lambda_{MSE}$ are important to obtain good results. In particular, we observed that $\lambda_{MSE}$ plays the most important role which is associated with the quality of the implicit networks and monitors how close the output of the network should be to the measurement.  For a larger noise, we should make the parameter $\lambda_{MSE}$ smaller, otherwise, we may overfit. In any case, these are hyperparameters, and we need to run a couple of experiments to find good values of them. In our cubic oscillatory example, we run the experiments for different $\lambda_{MSE}$, (e.g, $\{1,10,100,1000\}$) by keeping other parameters to $1.0$. We report the best results among the considered case. A detailed investigation needs to be carried out to determine the effect of each of these on the performance of the approach.
>
> * We have not done hyper-parameter tuning much about the network and learning rate.  One can indeed perform a hyperparameter tuning to determine the best architecture but it would come at additional computational costs.
>
> * In our revised paper, we have focused on only one example and have done a detailed discussion by comparing it with Neural ODE due to limited space. For learning the vector field, we used a residual-type network which is more robust to the number of residual blocks due to residual connections, and for the implicit network, we used a simple multi-layer perceptron with $\sin$ activation function. Since our goal is to learn a model-defining vector field, we do not employ RNN. We have performed a comparison with neural ODE which is one of the the-state-of-the art methods to learn ODEs. Please have a look at the updated numerical section in the paper.

---

### Decision · Program_Chairs · 2021-10-16

**Decision:**

Accept (Poster)

**Comment:**

Reviewers have recommended the acceptance of this article.